# Single Blinded Study on the Feasibility of Decontaminating LA-MRSA in Pig Compartments under Routine Conditions

**DOI:** 10.3390/antibiotics9040141

**Published:** 2020-03-26

**Authors:** Iris Kobusch, Hannah Müller, Alexander Mellmann, Robin Köck, Marc Boelhauve

**Affiliations:** 1South Westphalia University of Applied Sciences, 59494 Soest, Germany; kobusch.iris@fh-swf.de (I.K.); mueller.hannah@fh-swf.de (H.M.); 2Institute of Hygiene, University of Münster, 48149 Münster, Germany; Alexander.Mellmann@ukmuenster.de (A.M.); r.koeck@drk-kliniken-berlin.de (R.K.); 3Institute of Hygiene, DRK Kliniken Berlin, 14050 Berlin, Germany

**Keywords:** LA-MRSA, decontamination, standard cleaning and disinfection, pig farming

## Abstract

In countries with intensive pig husbandry in stables, the prevalence of livestock-associated (LA) methicillin-resistant *Staphylococcus aureus* (MRSA) on such farms has remained high in the last few years or has also further increased. Simple measures to reduce the LA-MRSA among pigs have not yet been successfully implemented. Earlier publications showed a decontamination of LA-MRSA was only possible with great effort. The aim of this study is to determine the suitability of routine cleaning and disinfection (C&D) for adequate LA-MRSA decontamination. For this purpose, at least 115 locations in a piglet-rearing compartment were examined before and after cleaning and disinfection. The sample locations were stratified according to accessibility for pigs and the difficulty of cleaning. The cleaning work was carried out routinely by farm employees, who were not informed about the sampling (single blinded). While before cleaning and disinfection, 85% of the samples from the surfaces were LA-MRSA positive, while only 2% were positive thereafter. All LA-MRSA-positive samples after cleaning and disinfection were outside the animal area. Air samples also showed no LA-MRSA after cleaning and disinfection. Conclusion: In well-managed livestock farms, decontamination of the LA-MRSA barn is quite possible; after C&D no LA-MRSA was detectable at animal height.

## 1. Introduction

Numerous studies have shown the occurrence of methicillin-resistant *Staphylococcus aureus* (MRSA) in livestock (LA-MRSA), especially in pigs [1]). For many years, Germany has been one of the countries with a high detection rate for LA-MRSA in pig farming [2,3]. It has also been widely discussed whether the use of antibiotics in livestock is an important source for the continuous spread of LA-MRSA in animals and humans [4]. Pig farmers can be directly affected by LA-MRSA prevalence in their own livestock, e.g., they are frequently (up to 80% of pig farmers) colonized by working continuously in air containing LA-MRSA and even become infected under unfavourable conditions [5] reviewed in [6]. Since 2006, the influence of LA-MRSA colonization caused by contact with livestock on the epidemiology of MRSA in hospitals located in rural areas and on human infections was clearly demonstrated [7,8]. A long-term decolonization of persons related to livestock farming has so far been unsuccessful, as these persons have been repeatedly recolonized if LA-MRSA was still detectable in the stable [6,9].

In addition to the discussion about the role of antibiotics in animals, the level of risk for non-agriculturally associated persons (“normal” population) of consuming meat from such LA-MRSA-positive animal husbandry is discussed differently [10,11], as seen in an overview by [12].

As a result, control points were sought to limit the transmission of resistant pathogens “from stable to table” and interventions to reduce the colonization of livestock populations were demanded. To date, only the intensive culling of LA-MRSA-positive livestock in Norway has proven to be successful [13]—however, only for a short time. This opportunity is neither possible in Germany (as it is a transit country in Europe) nor has the short-term success in Norway been convincing and socially accepted. Due to the high rate of live animal transport with pigs [14] and the not fully implemented hygiene barrier measures (e.g., separation of LA-MRSA-populated humans in animal husbandry from LA-MRSA-negative animals) in Germany [12], a consistent establishment of LA-MRSA-negative herds is difficult. Even the LA-MRSA-free status in Norway could not be maintained by interrupting hygiene barrier measures [13]. In addition, in large parts of Europe and the world culling actions would be viewed critically by the population, since in recent years a greater interest in animal welfare oriented animal husbandry has grown [15].

LA-MRSA bacteria are not only detected in the nares of animals, but also on surfaces that are in contact with the animals, such as stable walls and equipment. LA-MRSA can also be detected in air and dust samples [16,17].

However, the consistent implementation of infection control measures to reduce the prevalence of LA-MRSA is a challenge in pig production, because the number of animals per farm is sometimes very high, and the entire farm can rarely be described as “animal-free” at any given time [17]. The decontamination of livestock farms could only be carried out insufficiently so far, since the animals repeatedly served as a source for environmental re-contamination [9,18]. Therefore, it can be doubted that measures from human medicine can be modified directly and thus successfully implemented in animal husbandries [4]. For example, individual decolonization procedures for farm animals cannot be applied and financed.

Nevertheless, the remaining bacteria should be removed by cleaning and disinfection (C&D) after LA-MRSA-positive animals have been stabled out. So far, no resistance of LA-MRSA to commonly used disinfectants such as peracetic acid and quaternary ammonium compounds is known [19]. Previous investigations into the decontamination of LA-MRSA-positive pens have mostly been carried out with a high level of human or mechanical commitment, including partial renovation of the facilities [17,20].

This study focuses on whether the standard cleaning and disinfection methods under practical conditions used in pig farms could also be sufficient to provide decontamination of LA-MRSA.

## 2. Results

Within three repetitions, a total of 363 environmental samples were evaluated before cleaning and 363 samples after C&D. Directly after housing-in, 44 environmental samples in the bay and 58 nasal swabs of weaned piglets were collected. In week 1 there were 44 environmental samples and 57 nasal swabs, in week seven 44 environmental samples and 51 nasal swabs. The number of selected piglets decreased due to relocation to other compartments (*n* = 6), one piglet died. In addition, 12 twelve air samples and 18 water samples were taken. Of these 1010 samples, 481 were LA-MRSA-positive.

### 2.1. Environmental Analysis around C&D

Of the environmental samples, a total of 71% were LA-MRSA-positive before C&D (without animals inside the compartment), with increased positive samples inside the animals’ range.

Around 80% of the samples in the animals’ ranges were LA-MRSA-positive, whereas outside this ranges only 65.7% positive samples were found (*p* = 0.004) (Figure 1).

The air samples as well as the water samples taken before cleaning the compartment were all LA-MRSA-negative.

The prevalence of LA-MRSA-positive samples before C&D was slightly higher for the easy-to-clean areas (73.9%) than for the difficult to clean areas (70%) (*p* = 0.48) (Figure 2). After C&D, LA-MRSA was detectable on 3.3% (easy to clean areas) and 2.1% (difficult to clean areas) of the sampled surfaces, respectively (Figure 2, *p* = 0.49). The additional samples taken in repetitions II and III (each *n* = 7) were all MRSA-negative after C&D.

Positive samples were also identified in repetition I on the ground of the central corridor (outside the compartment) and in one of the supply air ducts (outside the animals’ range). In repetition II, one positive sample was found on and behind a water pipe as well as between the slatted floor elements (not accessible directly to animals). Moreover, in the last repetition there were four positive samples after C&D (on and behind a water pipe, in a corner of the compartment walls, feeding valve (outside the range of the animals).

Differences in LA-MRSA detection according to C&D (Table 1) were not observed in the different materials collected during sampling (*p* = 0.23).

Air and water samples taken after C&D were all MRSA-negative. The spa types t011, t034 and t8616 were found in the compartments. These spa types can be categorized as LA-MRSA CC398.

### 2.2. Analysis of Piglet Weaning Phase

Sampling during the rearing phase documented the development of LA-MRSA contamination of the compartment’s surfaces and piglets during the seven weeks following housing-in. From housing in to week 7 there were increasing numbers of LA-MRSA-positive samples from surfaces (1.7% to 83.7%, *p* < 0.001) (Figure 3).

Of the sampled pigs nasal swabs showed a rate of 71.7% LA-MRSA -positive animals at the day of housing-in. Piglets were partially (50% [repetition I], 65% [repetition II]) or complete LA-MRSA-positive [repetition III] at the time of housing. In week 1 and 7 all tested piglets were colonized by LA-MRSA.

Air sampling detected LA-MRSA already within 30 minutes after housing of the previously LA-MRSA-negative stables. After that time, the air samples remained LA-MRSA positive until the next C&D.

## 3. Discussion

The ongoing high prevalence of LA-MRSA in pig-rearing countries with intensive animal husbandry leads to an increase of infections with these pathogens in directly exposed persons (e.g., sepsis, soft tissue infection) [7]. Previous measures have not been able to reduce the prevalence of LA-MRSA among pigs [16,17]. As an animal-associated bacterium that colonises the nasopharynx, it could be assumed that LA-MRSA is mainly found in areas with direct animal contact [16]. On the other hand, LA-MRSA could be found more frequently in the air within and in the area around stables [17,21]. Our results support the hypothesis that the environment in the stable is predominantly contaminated in animals’ range rather than at other locations (Figure 1). However, before C&D, 80% of the samples were LA-MRSA-positive in the range of the animals and 66% of the samples outside of the range of the the animals. LA-MRSA detection was not limited to areas that are very difficult to clean and disinfect due to constructional reasons (e.g., inside of the bay partition wall) or where recontamination occurs due to unfavourable installation in the compartment (e.g., cables). The considerable number of positive samples found outside the animals’ range shows that these surfaces were reached, e.g., by contaminated dust. The various materials sampled showed no differences based on the different surface textures and thus also accessibility for cleaning the materials (Table 1). It should be mentioned here that the disinfectant used was selected on the basis of an efficacy analysis for bacterial pathogens in livestock stables (disinfectant test of the German Veterinary Medical Society). There are also preparations or active substances listed which have only a limited or no effect against bacteria and are effective against viruses or parasites, for example.

The challenge of LA-MRSA reduction remains in the fact that LA-MRSA-negative animal populations have to be found or built up and the stable environment has to be prepared in a way that prevents the LA-MRSA acquisition of negative animals. The basis of the latter challenge already exists with the technology and working methods available in the pig-rearing farms [22,23]. The aim of the study was to show the possibility to create LA-MRSA-free compartments (stables) by C&D in a previously contaminated environment. LA-MRSA-positive sampling areas could be reduced to 2% (areas difficult to clean) or 3% (areas easy to clean), respectively (Figure 2). It was apparently irrelevant whether these areas were classified as good or difficult to clean.

The reduction of LA-MRSA detection by standard C&D demonstrates under practical conditions that the existing techniques are suitable to effectively curb LA-MRSA from the environment. In this study, after C&D, LA-MRSA-positive samples could only be found in areas that could not be reached by subsequently housed animals. In the area that animals could reach during housing, no LA-MRSA could be found in all three replicates. This leads to the assumption that the importance of the C&D carried out seems to be very high among the persons not informed about the investigations. The selection of this company for this investigation was based on the rather high level of C&D we had determined in advance. The samples were taken at times when none of the farm employees were still on the farm, and only the farmer was present. The cleaning person was informed by us in the presence of the farmer after the evaluation of the three repetitions and confirmed that he had no knowledge of the examinations. The quite well executed C&D with the high reduction results for LA-MRSA in three repetitions show the general suitability of the standard C&D in well-managed farms. For this study, it was important to perform the decontamination in a practical framework in order to ensure the transfer to routine agricultural use. In addition, the LA-MRSA positive areas were not identical and could not be recognized repeatedly during each rearing phase. All areas, which were determined LA-MRSA positive after C&D, can most likely get rid of it. It can be assumed that this contamination was caused by the C&D itself and does not represent a subsequent contamination of these areas by boots or clothing, as these contamination were found in areas that were difficult to reach through them (e.g., supply air ducts, between the slatted floor elements).This simple blinded study also indicates the need for intensive training for good C&D, so that LA-MRSA negative animals could remain negative. The basic prerequisite for the removal of LA-MRSA by C&D is not only the good professional training of the cleaning staff, but also the appropriate selection of a suitable disinfectant, the right application technique (e.g., high-pressure cleaner with suitable disinfection application nozzle) and a high level of implementation of hygiene barriers (avoidance of cross-contamination). In this investigated animal husbandry, all these parameters were fulfilled. Unfortunately, it was not possible to identify a piglet producer keeping LA-MRSA negative piglets prior to the study. Therefore, the piglets were examined for assurance, although previous investigations in this barn had shown that these animals are colonized with LA-MRSA in a very variable manner at the beginning of the flatdeck phase [24]. Decolonization of the piglets was not the aim of this study and was not expected based on previous studies [9,16].

LA-MRSA detection on the compartment corridor or central corridor outside the compartment indicates that contamination may have occurred from areas of the barn that still contained LA-MRSA-positive pigs. This situation also shows the difficulty of a consistent decontamination of entire pig houses, since at least one compartment is normally still occupied with pigs, while the others are cleaned and disinfected. In the transitional phase of LA-MRSA decontamination of entire pigsties, which cannot be completely cleaned and disinfected, it would be useful to establish a consistent hygiene barrier. For the housing of LA-MRSA-negative animals, a consistent implementation of barrier measures to avoid LA-MRSA carryover from positive to already negative areas is urgently required. This includes changing boots and clothing, washing hands and wearing respiratory masks until the entire farm, including people, is LA-MRSA negative. It would be desirable to study this C&D quality in farms that have a LA-MRSA-negative piglet reference. Unfortunately, when the experimental design was created, no farm could be found that both carried out a relatively good C&D and had LA-MRSA-negative piglets.

Air and water samples were all MRSA-negative, when the animals were not in the compartment (this applies to the sampling before and after C&D). As soon as new animals were housed-in, the MRSA status of the environment changed within days (Figure 3). Individual LA-MRSA-negative animals were tested positive after rehousing. The results of the air samples in the occupied barn illustrate the increased LA-MRSA pressure of the LA-MRSA-positive animals. This underlines the role of the animals as LA-MRSA reservoirs and as main disseminators of LA-MRSA, which is also shown in a study by Bangerter et al. [25]. Other studies also describe the stable’s recontamination by LA-MRSA-positive animals [26,27]. Plentinckx et al. [18] detected the relationship between the environmental and the animals’ prevalence. In this study [18] sows were washed followed by a disinfection of the skin at the time of bringing them to nursery barns. The prevalence of LA-MRSA on the sows’ and piglets’ skin was reduced significantly but there a long-term effect could not be achieved [18].

## 4. Materials and Methods

### 4.1. Aims of the Study

The investigation was divided in two parts: First, we extensively assessed environmental contamination with LA-MRSA in a freshly cleaned animal barn and second, we analyzed the possibility of decontaminating a previously LA-MRSA-positive barn by standard cleaning and disinfection under practical conditions.

### 4.2. Organisation of the Farm and/or Stable

This research was carried out in a conventional swine farm with sow husbandry and piglet production including weaning pigs (790 sow places, 4200 weaning pig places). This farm was selected as it has consistently shown very good results in previous studies on the quality of the C&D carried out. In addition, it was also possible to carry out a single-blinded study in which the performing cleaner was not informed about the subsequent sampling after cleaning and after disinfection. Furthermore, the sampling took place at a time when the cleaning person had already left the farm.

Normally, piglets are born in a two-week interval followed by a suckling phase of 24 days. Piglets are individually marked with ear tags at the end of the suckling phase as per usual in the farm. After weaning, the rearing phase comprises about 50 days and is performed in special compartments with standard flatted floor, a ventilation/heating system and liquid feeding. These flatdecks were subject of this work. The size of the researched compartments was identical: 2.7 m high, 8.8 m wide, and 16.5 m long (on average 0.35 m^2^ per animal), with five pens of the same size each on the right and left side of a control corridor. The material of the floor, ceiling, walls and parts of the enrichment tools (e.g., balls, tubes) consisted of plastic or had a plastic surface. Feeding trough, nipple drinkers and other parts of the enrichment tools (e.g., link chains) were made of stainless steel or iron. In each pen, 40 weaners with an average weight of 6 to 8 kg were introduced at the day of weaning.

### 4.3. Cleaning and Disinfection (C&D)

After the previous animal group had moved out, the compartment was empty for two to five days. Farm employees cleaned and disinfected the weaning pig compartments as usual and thus under standard conditions. This procedure included the manual removal of the partially dried manure, a soaking of the residual dirt with water over several hours with a soaking system, a pre-cleaning of the compartment with a high pressure cleaner, followed by a foam cleaner phase with sodium hydroxide (product: Einweichschaum, BestFarm, Ascheberg, Germany) and an intensive cleaning phase with a high pressure cleaner. After a drying time of about 18 h, all surfaces up to a height of 2 m were treated with a foam disinfectant (product: hydrogen peroxide and peracetic acid, Sorgene 5, BASF, Germany). These work instructions were standardized through standard operation procedures with detailed list of preparations, concentrations and equipment to be used and employees were instructed and trained accordingly by the farm owner.

The piglets were usually stabled the next day after disinfection.

### 4.4. Sampling Surfaces around Cleaning and Disinfection

The study was conducted in three repetitions from February to November 2016. Environmental contamination of surfaces, air and water with MRSA was determined before (without animals inside) and after C&D at defined positions in a compartment (Table 2).

Sampling took place in the entire test compartment, with certain bays and locations being selected without informing the persons responsible for C&D (single-blinded). The samples were taken outside the working times of the cleaning personnel. A fixed sampling plan was established for all three replicates before sampling (*n* = 124), in repetition I *n* = 124 samples were taken, nine samples could not be evaluated due to contamination on the culture media and were therefore removed from further analysis. In advance, 124 locations of surfaces (each in repetitions II and III, 115 in repetition I) were identified, which were either easy or difficult to clean and inside or outside the range of the animals (Table 2). Areas that are easy to clean were defined as locations that can be reached easily and were obviously visible during standard C&D. Examples of areas that were easy to clean are the floor, walls, windows, the ceiling or feed troughs. Here, 63 (62 in repetition I) samples were taken, 28 of them with direct animal contact. For example, areas that were difficult to clean included locations such as in gaps or under objects and could not be reached without further measures [61 samples (55 in repetition I); 31 of them with direct animal contact]. Gaps between the flatted floor, under the feeding troughs, behind water pipes, between cables or the lamps are examples of areas that are difficult to clean. The different material surfaces, such as stainless steel, plastic, metal or concrete, were also recorded during sampling in the four categories (Table 2). Some samples of surfaces were also collected outside on the central corridor in front of the compartment (*n* = 12, *n* = 6 before and *n* = 6 after C&D).

In the following figures the test compartment and the sampling locations were described, whereby the lower (<2 m height) and upper (>2 m height) areas of the compartment are shown (Figure 4 and Figure 5). The lower area included places that were easy to reach by animals (animals’ range, e.g., floor in the bays, feed troughs, nipple drinkers) (Figure 4). The upper area also involved the main feeding pipe, the supply air duct and the exhaust air duct (outside the range of the animals, e.g., the floor of the control corridor, the ceiling including the elements of the ventilation system) (Figure 5).

Swabs with Amies transport medium (VWR) were used for sampling of the surfaces. Depending on the location of the sampled area, defined areas (20 cm^2^, 4 × 5 cm) or distances (10 cm × 2 cm) were sampled.

### 4.5. Sampling Surfaces during Rearing Phase

For the surface sampling during the rearing phase, 12 (20 in repetition I) surfaces inside the animals’ range were chosen. Samples from surfaces and animals were collected at three points in time during the rearing phase: on the day of the housing (day of weaning), week 1 and week 7 after weaning. Sampling procedures were similar as described before.

### 4.6. Sampling Animals

Nasal MRSA carriage of 20 (18 in repetition III) randomly selected piglets by individual ear tag number per repetition (two animals per bay) was analyzed directly after housing, in week 1 and 7 after housing in at parallel time to the surface sampling as described before. The same two animals per pen were sampled at each time. The nasal atria of both nostrils were sampled with swabs with Amies transport medium (VWR, Langenfeld, Germany). The swab was taken in rotating movements without touching the outside of the snout. Pigs were identified with an electronic ear tag (MS Quick Transponder FDX, MS Schippers, Bladel, Netherlands) for individual analysis.

### 4.7. Sampling Air and Water

Two air samples were each taken before and after C&D by an air sampler (MicroBio MB2, Cantium Scientific, Dartford, UK) by aspirating 100 liters of air within one minute. Air samples are drawn over a chromID™ MRSA SMART Agar (bioMérieux, Nurtingen, Germany) and immediately closed in the barn until further incubation at the laboratory. Furthermore, three water samples from the externally disinfected nipple drinkers were each taken before and after C&D by filling 250 mL into a sterilized glass bottle and cooling until examination. All samples (from surfaces, animals, air and water) were examined directly in the nearby laboratory at the same day of collection.

### 4.8. Bacterial Culturing and Spa Typing

For microbiological analysis, the nasal and environmental swabs were transferred in to 9 mL Mueller Hinton broth + 6.5 % NaCl (Mediaproducts BV, Groningen, Netherlands) and incubated at 37 °C for 18 ± 2 h for enrichment of staphylococci. From the Mueller Hinton broth + 6.5 % NaCl 500 µL were added to 5 mL Tryptone soya broth + cefoxitin/aztreonam (Mediaproducts BV, Groningen, Netherlands) and grown over 18 ± 2 h at 37 °C for MRSA-enrichment. Subsequently 10 µL of the enriched cultures were inoculated on chromID™ MRSA SMART Agar (bioMérieux, Nurtingen, Germany) and incubated at 37 °C for 24 h. Plates of air samples were incubated as specified by the manufacturer. Typical colonies grown on chromID™ MRSA SMART Agar (bioMérieux, Nurtingen, Germany) were selected. Spa typing has been performed before following a previously described protocol [28]. Briefly, we amplified an rpoB gene section (899 bp) with Staphylococcusspecific primers and sequenced it subsequently. For further analysis, nucleotides 1444-1928 of the rpoB gene were used. Air samples on chromID™ MRSA SMART Agar (bioMérieux, Nurtingen, Germany) were incubated under same conditions. For the microbiological analysis of the water samples, 1 mL of the water sample was added to chromID™ MRSA SMART Agar (bioMérieux, Nurtingen, Germany) and incubated as previously described.

### 4.9. Statistical Analysis

For analysis of metric variables, calculated mean values were tested for statistical significance between groups by chi-square tests. For statistical analysis, the software GraphPad Software (version 6, San Diego, CA, USA) was used. Percentage data was analyzed without transformation and after being subjected to the arcsine transformation to correct for problems of non-normality associated with analysis of percentage data.

## 5. Conclusions

This study concerns the decontamination of LA-MRSA in stables for weaning pigs, as well as the detection of LA-MRSA hotspots in stables before and after standard C&D. In addition, the colonization of the surfaces over time is considered. In this study environmental LA-MRSA was considerably reduced by the process of C&D under standard conditions. All analyzed areas can most likely get rid of LA-MRSA by standard C&D. A rapid recontamination during the weaning period was due to LA-MRSA-positive animals, so the animals serve as main disseminators for LA-MRSA as previously reported.

In order to establish LA-MRSA-free animal husbandry, LA-MRSA negative animals must be transferred to an LA-MRSA-free barn. In addition, possible sources of contamination (people, other animals) would have to be decolonized or strictly separated in parallel. Complete decontamination of barns would be possible by training the staff responsible for C&D. However, this would only be considered sensible in combination with complete decontamination of a pig holding. The importance of hygienic barrier measures would also be understandable to all persons involved and could be implemented more consistently.

The examination of the permanent LA-MRSA negative barn over a complete animal husbandry period will only be possible when completely LA-MRSA negative animals are available by measures or methods not yet developed.

## Figures and Tables

**Figure 1 antibiotics-09-00141-f001:**
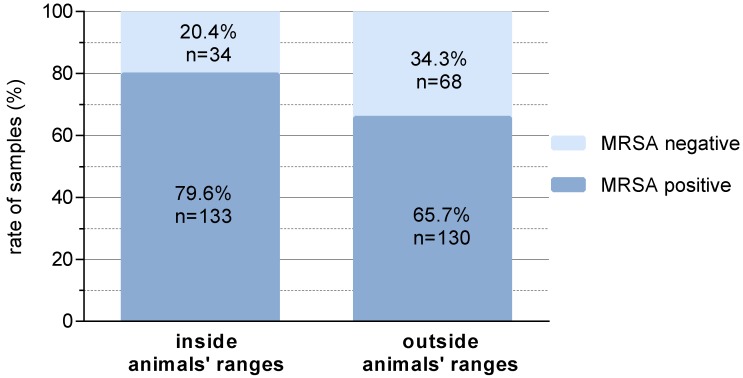
Prevalence of livestock-associated methicillin-resistant *Staphylococcus aureus* (LA-MRSA) before cleaning and disinfection (C&D) inside and outside animals’ ranges.

**Figure 2 antibiotics-09-00141-f002:**
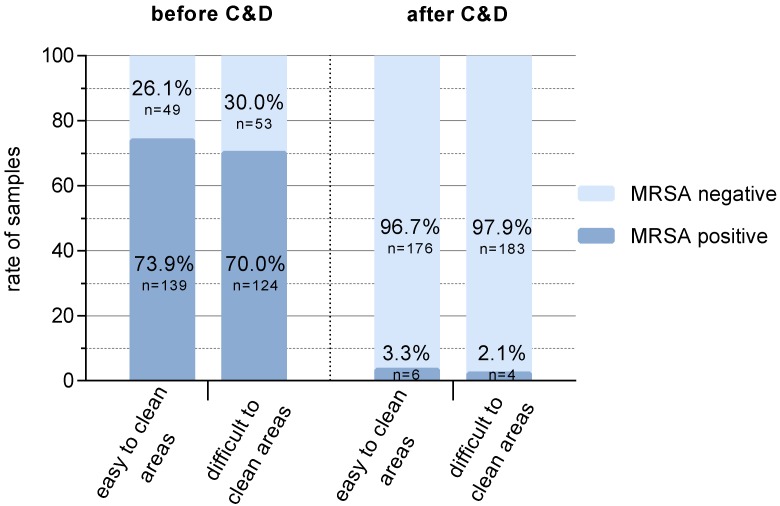
Prevalence of LA-MRSA before and after C&D depending on the difficulty of cleaning of the weaning pig compartments. Differences between easy and difficult to clean areas were not significant.

**Figure 3 antibiotics-09-00141-f003:**
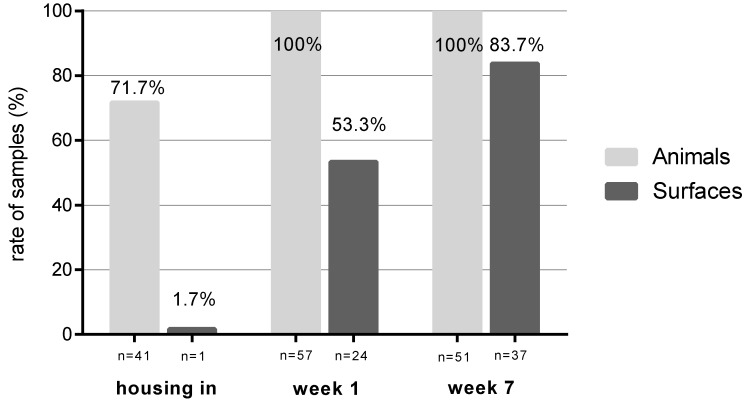
Rate of LA-MRSA prevalence at three times of sampling during the weaning period.

**Figure 4 antibiotics-09-00141-f004:**
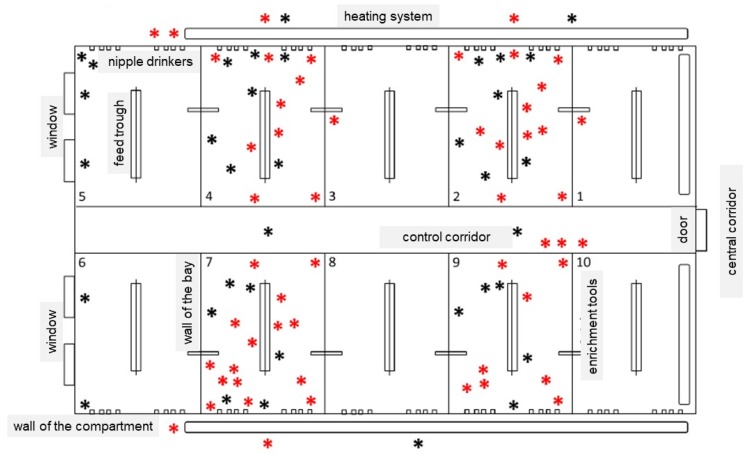
Schematic overview of the sampling locations in the lower part of the test compartment (<2 m height). ***** Black dots: easy to clean areas. ***** Red dots: difficult to clean areas.

**Figure 5 antibiotics-09-00141-f005:**
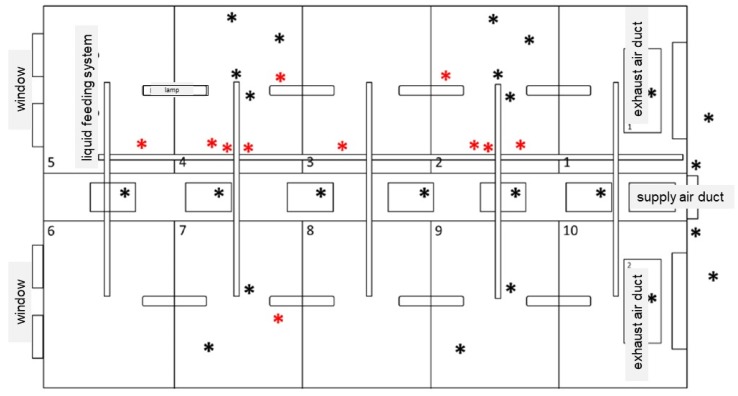
Schematic overview of the sampling locations in the upper part of the test compartment (>2 m height). ***** Black dots: easy to clean areas. ***** Red dots: difficult to clean areas.

**Table 1 antibiotics-09-00141-t001:** Materials sampled and LA-MRSA detection depending on C&D.

Surfaces/Material	Before C&D			After C&D		
	MRSA-Positive	Total Number	%	MRSA-Positive	Total Number	%
Plastic	128	175	73	5	176	3
Metal	32	48	67	0	50	0
Stainless steel	62	86	72	2	87	2
Concrete	30	40	75	3	41	7

**Table 2 antibiotics-09-00141-t002:** Sampling locations.

	Easy to Clean areas	Difficult to Clean Areas
in animals’ range	area	*n* =	area	*n* =
floor in the bay	12	between slatted floor elements	12
wall of the bay	12	under and corners wall of bay	24
wall of the compartment (<2 m)	12	under feed trough	12
wall over feed trough	12	nipple drinkers	9
corner walls of compartment	9	in mount for feeding pipes	9
in feed trough	12	behind water and feeding pipes (<2 m)	18
corner slatted floor and wall	6	enrichment material	9
on water and feeding pipes (<2 m)	9		
			optional samples: hollow space in wall of bay	12
outside animals’ range	area	*n* =	area	*n* =
windows	9	on and in heating pipe	18
wall of the compartment (>2 m)	9	behind water and feeding pipes (>2 m)	18
door compartment	6	on main feeding pipe	6
feeding pipes (>2 m)	18	between cables feeding pipe	9
on heating pipe	9	lamps	9
supply air duct	18	under slatted floor	30
exhaust air duct	6		
ceiling	12		
floor central corridor	9

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
