# Peer review of "Single Blinded Study on the Feasibility of Decontaminating LA-MRSA in Pig Compartments under Routine Conditions"

_antibiotics, 2020, doi:10.3390/antibiotics9040141_

Round 1

Reviewer 1 Report

This is a well designed study addressing an important problem. The results show a positive effect of the implemented regimen.

The paper could be improved by addressing a few points:

  • There are not data on the molecular typing of isolates, in method and results. How LA-MRSA was confirmed? 
  • What were the findings and effects on MRSA in general, and only specific LA lineages
  • The broader One Health context needs to be discussed including a risk assessment to human health 
  • The authors should discuss the generalisability of the their approach

Reviewer 2 Report

The topic of this paper is interesting and it only requires minor changes.

Single blinded study on the feasibility of decontaminating LA-MRSA in pig compartments under practical conditions

The topic of this paper is very interesting.

TITLE

I suggest this title: "Single blinded study on the feasibility of decontaminating LA-MRSA in pig compartments under routine conditions"

ABSTRACT

Line 14, it says "...professional pig husbandry...". Do you mean intensive pig husbandry?

Line 18, it says "Methods: The aim of this study...". In my opinion, you should split aim and methods.

ENGLISH LANGUAGE

In my opinion it is very good.

INTRODUCTION

Lines 48-50, it says " This opportunity is neither possible in Germany (transit country in Europe) nor has the short-term success in Norway been convincing., nor is it socially accepted". Please give more information and reason it.

Line 67, split this paragraph; new paragraph from “This study…”

RESULTS

Line 84, Figure 3 is the first one; please change numbering along the paper.

Lines 105-106, it says " The spa types t011, t034 and t8616 were found in the compartments." There is not any other mention to this in the paper. Please explain it in methodology.

Line 119, what is the percentage of positive samples?

DISCUSSION

Lines 124-125, it says "Previous measures have not been able to reduce the prevalence of LA-MRSA among pigs". I miss any reference.

Lines 125-126, it says "As an animal-associated bacterium that colonises the nasopharynx, it could be assumed that LA-MRSA is mainly found in areas with direct animal contact". I miss any reference.

Line 136; please, mention table 2.

Lines 140-141, it says "The basis of the latter challenge already exists with the technology and working methods available in the pig-rearing farms". I miss any reference.

Lines 148-149, it says "In this study LA-MRSA detection was limited to areas that the animals could not reach". However, there are areas inside animals' ranges. What do you mean?

Lines 149-150, it says "This leads to the assumption that the importance of the C&D carried out seems to be very high among the persons not informed about the investigations". I do not agree with this assertion. There was not any evaluation of these persons.

Lines 156-158, it says "It can be assumed that this contamination was caused by the C&D itself and does not represent a subsequent contamination of the barn, as these were found in areas that could not be dragged away by boots or clothing". I do not agree with this assertion. Please, rewrite it.

Lines 161-163, it says "Therefore, the piglets were examined for assurance, although previous investigations in this barn had shown that these animals are colonized with LA-MRSA in a very variable manner at the beginning of the flatdeck phase". I miss any references.

Lines 163-164, it says "Decolonization of the piglets was not the aim of this study and was not expected based on previous studies [9,16]". How or why could it be expected? What do you mean?

Lines 167-169, it says "This situation also shows the difficulty of a consistent decontamination of entire pig houses, since usually at least one compartment is still occupied with pigs, while the others are cleaned and disinfected". You could suggest any solution. Boot change system? Foot baths?

Line 173, it says R&D. Do you mean C&D?

Lines 174-175, it says "In a total of more than 42 piglet farms investigated, none was identified with a fully LA MRSA-negative status (data not shown)". Delete this sentence or give more information.

Lines 184-185, it says "Plentinckx and co-workers detected the relationship between the environmental and the animals’ prevalence [16]". Please, write " Plentinckx et al. [16] detected ..."

Lines 185-188, it says "Here sows were washed followed by a disinfection of the skin at the time of bringing them to nursery barns. The prevalence of LA-MRSA on the sows’ and piglets’ skin was reduced significantly but there a long-term effect could not be achieved [16]". Do you mean in this paper or in [16]. Please, clarify it.

MATERIAL AND METHODS

Lines 197-199, it says "This farm was selected as it has consistently shown very good results in previous studies on the quality of the C&D carried out". I miss any reference of these previous studies.

Lines 202-203, it says "These conditions were not met in any other pig holding outside this holding in the run-up to this study". Please, delete this sentence.

Lines 202-203, it says "These compartments (including the rearing phase) were subject of this work". What do you mean? I understand you have sampled and studied a post-weaning room and weaners. Please, clarify and rewrite it.

Line 224, it says "These work instructions were standardized...". Could you detail these?

Line 228 (and line 71), it says "The study was conducted in three repetitions ...". Could you detail these? Number of samples of each repetition...

Line 252 and figures 1 and 2. In my opinion these figures are not necessary.

Line 269, it says "...animals were collected at three points...". It is "three moments".

CONCLUSIONS

Lines 312-315, it says "Seventy-one percent of environmental samples (surfaces of the stable) were LA-MRSA-positive before C&D. There were increased positive samples inside animals’ range (80% inside, 66% outside). After C&D, 2 to 3% of the surface samples were LA-MRSA-positive". These are results; these should be conclusions. Please, delete it.

Lines 318-319, it says "From housing in to housing out there was detected an increasing number of positive surface samples from 2% to 84%". These are results; these should be conclusions. Please, delete it.
